# Visual Quality Assessment of Urban Scenes with the Contemplative Landscape Model: Evidence from a Compact City Downtown Core

**Hou Yanru [1], Mahyar Masoudi [1], Agnieszka Chadala [2] and Agnieszka Olszewska-Guizzo [2,3,\*]**

[1] Department of Architecture, School of Design and Environment, National University of Singapore, 8 Architecture Drive, Singapore 117356, Singapore; e0305534@u.nus.edu (H.Y.); mahyar@u.nus.edu (M.M.)

[2] NeuroLandscape Foundation, Suwalska 8/78, 03-252 Warsaw, Poland; a.chadala@neurolandscape.org

[3] Institute for Health Innovation & Technology (iHealthtech), MD6, 14 Medical Drive, Singapore 117599, Singapore

\* Correspondence: a.o.guizzo@neurolandscape.org

**Abstract:** In the face of rapid urbanization and the growing burden of mental health disease, there is a need to design cities with consideration for human mental health and well-being. There is an emerging body of evidence on the importance of everyday environmental exposures regarding the mental health of city inhabitants. For example, contemplative landscapes, through a series of neuroscience experiments, were shown to trigger improved mood and restoration of attention. While the Contemplative Landscape Model (CLM) for scoring landscape views was applied to single images, its suitability was never tested for walking paths and areas with a diversity of viewpoints. This study aims to fill this gap using the high-density downtown of Singapore, also known as a "City in a Garden" for its advanced urban greening strategies, as a case study. In this study, 68 360° photos were taken along four popular walking paths every 20 m. A photo set of 204 items was created by extracting three view angles from each photo. Each of them was independently scored by three experts and average CLM scores for each view and path were obtained. The results were then fed into an open-source Quantum Geographic Information System (QGIS) for visualization. Cohen's kappa agreement between experts' scores was computed. The outcomes were mapped to facilitate the identification of the most contemplative viewpoints and paths. Moreover, specific contemplative landscape patterns have been distinguished and assessed allowing the recommendation of design strategies to improve the quality of viewpoints and paths. The inter-rater agreement reached substantial to perfect values. CLM is a reliable and suitable tool that enables the fine-grained assessment and improvement of the visual quality of the urban living environments with consideration of the mental health and well-being of urbanites. It can be used at a larger scale owing to 360° photos taken from the pedestrian's point of view. Utilizing spatially explicit maps in QGIS platforms enables a wider range of visualizations and allows for spatial patterns to be revealed that otherwise would have remained hidden. Our findings demonstrate the usefulness of our semi-automated method. Furthermore, given the high inter-rater agreement observed, we suggest that there is potential in developing fully automated methods.

**Keywords:** contemplative landscape; spatial analysis; mental health; urban; view

## 1. Introduction

Mental health issues including depression, anxiety, substance abuse and neurodegenerative diseases not only degrade people's quality of life, but also lead to serious economic losses [1,2]. Confronting these issues has become one of the major challenges of the contemporary world [3,4].

The COVID-19 pandemic has further exacerbated this challenge, with anxiety, uncertainty, and social isolation increasing on a global scale [5,6], even though scientists have not yet established the effect size of this phenomenon [7]. Improving mental health and well-being are no longer merely issues of medicine and the public health domain, but cut across several disciplines, and this intrinsically requires an integration of urban planning and design to produce healthier living environments.

Among the various determinants increasing the risk of mental disorders (e.g., genetic predispositions, individual behavior, habits, etc.), there are also environmental determinants related to the rapidly changing urban realm. The growing body of research shows that the everyday environmental sensory exposures can have a profound impact on the quality of life including effects on the risk of developing mental health disorders [8,9]. Urban environments characterized by high density built elements and infrastructure pose environmental challenges (traffic congestion, elevated temperatures, air and noise pollution) and are associated with increased burnout and stress [10]. This may explain the higher prevalence of mental health disease in urban as compared to rural environments [11]. On the other hand, nature exposure has been shown to induce psychological recovery (e.g., from stress and mental fatigue) [12–14], positive emotions [15,16], and to improve cognitive performance, memory, and creativity [17,18].

These insights and, more recently, findings about the contemplative landscapes [19,20], support the widespread use of urban greening solutions, such as the conservation of native vegetation, adding more greenery to the city overall, and incorporating elements of the natural environment into buildings and interior design.

## 1.1. Contemplative Landscape Model

The contemplative landscape model (CLM) developed by Olszewska-Guizzo provides the possibility to systematically assess the quality of landscape design, recognizing that each landscape view/scene has some contemplative value, determined by the aggregation of seven key-features, including: landscape layers, landform, vegetation, color & light, compatibility, archetypal elements, and the character of peace and silence [21]. Figure 1 illustrates the CLM with corresponding scores on a 1–6-point Likert scale and brief descriptions of each feature. CLM was developed and operationalized to enable expert-based evaluation of the contemplative value of various views and scenes with special consideration of the urban context, including urban green space (UGS) as built elements are important variables in the compatibility and character of peace and silence categories CLM is, to the best of our knowledge, the only visual assessment tool employing such a nuanced approach to landscape quality assessment, and which has been operationalised in order to be used in mental health research.

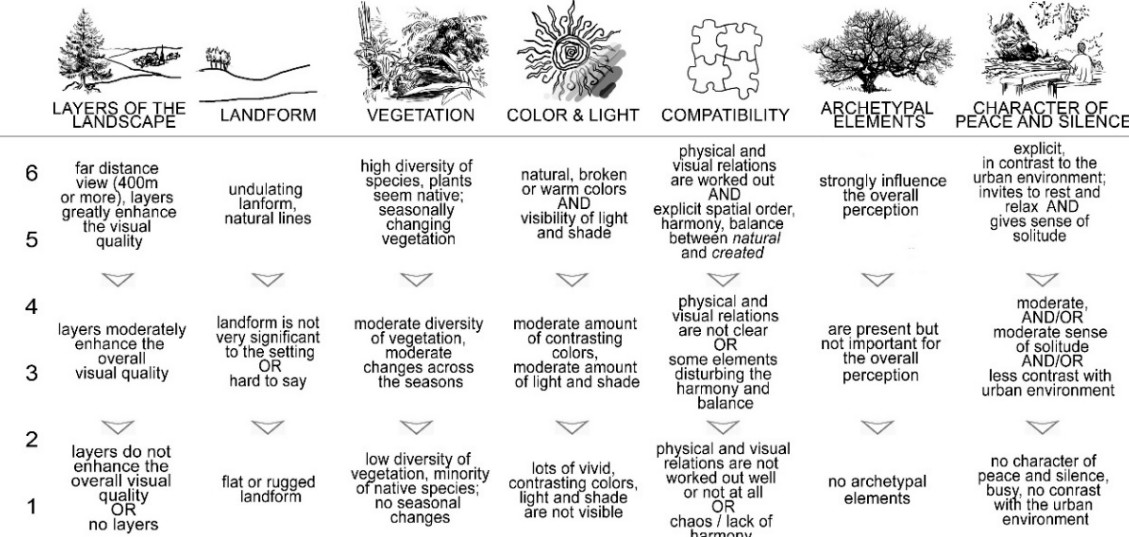

**Figure 1.** Simplified Contemplative Landscape Model for built landscape evaluation with 1 to 6-point scoring scale; adapted from [22].

It has been found through a series of neuroscience experiments that any view with dominant greenery seemed to induce momentary brainwave patterns associated with attention restoration, stress reduction, positive emotions and pleasantness, as compared to a resting state. However, for highly contemplative landscapes (CL score > 4.33 points), this pattern was significantly stronger when compared to landscape with relatively low contemplative (CL scores < 3.76 points) green scenes [19,22]. Correspondingly, preliminary findings suggest that passive in-situ exposure to highly contemplative landscapes (CL scores > 4.90 points) can trigger the brain response associated with positive approach and motivation [20]. The CLM can be used in-situ at selected viewpoints or remotely on a given photo taken from a landscape scene. Then, one or more experts assess each of the seven key components of the landscape individually by attributing a score from a 1–6-point scale, where 1 is the lowest and 6 is the highest possible score. The overall contemplative score can then be calculated by a simple average of the scores from seven key-features. Trained experts can spend around one minute to assess one view. Based on the results of the CLM evaluation, the most contemplative landscape views or sets of views can be easily identified from the given green space, and potentially protected and promoted. Furthermore, certain landscape design and maintenance activities can be recommended by the expert(s) to improve the existing CLM score.

### 1.2. Mental Health and Urban Greening in Singapore

Singapore is a high-density, highly urbanized compact city-state covering 721.5 km$^2$ with ~5.6M population and about 50% of green coverage [23]. Singapore is not an exception when it comes to the growing burden of mental health issues. The latest Singapore nationwide study shows that one in seven of the adult population in Singapore has experienced a mental disorder in their lifetime [24]. It is worth noting that the lifetime prevalence of all mental disorders will only increase, especially for generalized anxiety disorder and alcoholism. Studies have also shown that the age groups of 18–34 and 35–49 in Singapore are more likely to develop mental illness than children and the elderly [24].

At the same time, Singapore is considered an exemplar of the "Garden City" and its green efforts can be traced back to its first years as a sovereign state in the late 1960s [25]. One of the earliest initiatives was the Garden City vision of Lee Kuan Yew, the founding father and thought leader of the country who served as prime minister in the proceeding decades, to turn Singapore into a city of lush greenery with a clean environment. Subsequently, the Environmental Public Health Act was implemented in 1969 to strengthen Singapore's health legislation and raise public health standards. The Singapore Green Plan, released in May 1992, is the country's first formal plan to balance environmental and developmental needs [26].

Currently, the greening standards in Singapore seem to predominantly focus on the amount of green area and their proximity to residential places [27]. However, as studies are increasingly showing other attributes of UGS that affect their ecological performance, e.g., the level of fragmentation and geometrical complexity of green patches [23,28–30], several studies have pointed out the weak relationship between UGS quantity and mental health outcomes and have identified the need to consider other aspects, including the quality of greenery [31–33]. For example, seasonally changing diverse vegetation can invoke the cyclical nature of life [34], long distance views may increase feelings of personal freedom and mystery [35,36], and the presence of water or a single old tree may elicit an emotional response to a symbol of the collective unconscious [37], among other features. Moreover, in their research agenda, Frumkin and colleagues, concluded that "standard exposure measures are not grounded in the ecological elements most relevant to human health and wellbeing" [38]. For example, the quantity of nature is often measured using aerial photography or remote sensing techniques—such data offer little information on the quality of the landscape view from the point of view of an actual observer, nor does it focus on specific spatial attributes. More knowledge on the type/characteristics of visual scenery is crucial to understand potential contribution to human mental health [38–40]. Therefore, studies examining the relationships between green space and health should go beyond the mere amount of green space and include other variables that influence health outcomes.

### 1.3. Research Objectives

The goal of this study is to fill this gap in knowledge using the CLM to assess the visual quality of the scenes with consideration to the theory of environmental sensory exposures and their influence on the mental health and well-being of urbanites. We aim to test the CLM tool on the scale of urban walking routes, with larger set images derived from 360° photos taken from the point of view of a pedestrian. It is expected that the outcomes of this study can serve in multiple urban contexts, paving the way for more mental-health-aware urban planning and design, and to an improvement of the quality of every day environmental exposures.

## 2. Materials and Methods

### 2.1. Study Site Selection

With Singapore's growing burden of mental health problems in adults and its simultaneous widespread urban greening practices, the downtown urban core, commonly referred to as the central business district (CBD), was selected as the ideal testing site for the CLM tool (Figure 2a,b). As the main commercial area of the city-state of Singapore, the offices of innumerable companies and corporations are located here. According to the 2019 global Cigna 360 Well-Being Survey, Singaporeans are among the most stressed at work, globally, with 92% of working Singaporeans reporting to be stressed at work [41].

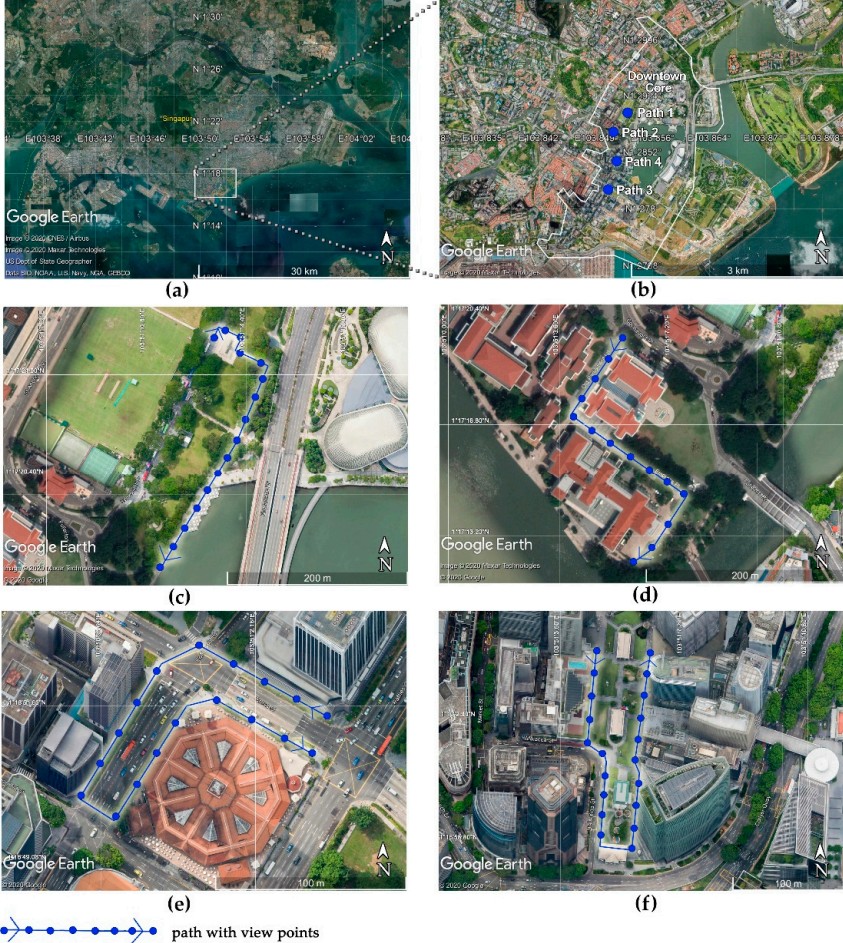

**Figure 2.** Study sites selection: (**a**) Singapore downtown core (**b**) 4 sample paths: (**c**) Path 1-Esplanade Park, (**d**) Path 2-Museum Area, (**e**) Path 3-Robinson Rd and (**f**) Path 4-Raffles Place Park.

As for the population data of the downtown core, the age distribution of people active in this area is mainly between 30–49 years old [42], which coincides with the age group mentioned above that

is predisposed to mental illness. Therefore, working professionals spend the majority of their week in this area [43], which means their environmental exposures are shaped with sensory stimulation from here. Even though the CBD is already full of green-blue solutions in the form of green buildings, green walls, street trees, and potted plants, as well as water features, according to development plans, it will continue to evolve to be more creative, connected, and vibrant also after work hours [43].

Therefore, assessing the contemplativeness of the landscape in this area to identify improvement areas may benefit its future development in regard to the mental health of Singapore residents interacting with the CBD.

Four popular walking paths were selected for this study based on the following criteria so that each represents different spatial composition (e.g., enclosed, focal) and viewpoints consisting of various landscape elements (e.g., flat lawn, water body, seating).

Path 1-Esplanade Park. The park was built in 1943 and is one of the oldest heritage parks in Singapore. The waterfront scenery and modern buildings surrounding Marina Bay can be seen from this path. It is one of the most popular and iconic parks in the downtown core of Singapore. There are different landscape elements on the preset study path, such as: tree-lined avenue, rest plaza, waterfront boardwalk, among others (Figure 2c).

Path 2-Museum Area. This area contains many famous exhibition venues, including the National Gallery, the Asian Civilization Museum and the Victoria Theatre. It is frequented by locals and tourists and has different landscape elements, such as historic buildings, edge of the park and an open plaza. The path (Figure 2d) passes through the buildings in the museum area, the rest area, and the main lawn area.

Path 3-Robison Rd. It is an important traffic corridor of the downtown core, with high-density traffic and business. As per Urban Redevelopment Authority (URA) planning, this road will have wider sidewalks, which means more space for vegetation and activities like al-fresco dining [43]. The research path (Figure 2e) is planned to start from Cross St, then enter Robinson Rd, and return in the opposite direction along Telok Ayer food market. The path is characterized by linear, focal views infringed by high-rise office buildings.

Path 4-Raffles Place Park. An enclosed park in the CBD's architectural complex, used by pedestrian employees from the nearby offices. According to the URA plan, Raffles Place Park "will be transformed into a vibrant focal point for the community, with enhanced design and programming" [43]. It serves as a precious green refuge in a dense "urban jungle", with open lawn and multiple seating options. The research route is a loop with a beginning and end at the metro (MRT) exit (Figure 2f).

*2.2. Photographic Data Collection*

Photos available in online resources like Google Street View, are for the most part, taken from the point of view of a driver, or to be even more precise, from the top of a car, which can be dramatically different from the point of view of a pedestrian. For that reason, we opted for a more accurate approach in view of our objectives, and took our own photos.

Photographic data collection took place between 1 and 30 January 2020, between 9 and 11 AM. 360° photos were taken in similar weather condition, at the shooting points pre-selected before going to the site with 20m distance between each other (Figure 2c–f). The photos were taken at 165cm above the ground, with a RICOH Theta Z1 360 Camera and pre-processed in Ricoh Theta Z1 Software UVC4K for Windows (Ricoh Company, Ltd.). Three viewpoints were extracted from each photo according to the 120° horizontal and 55° vertical limits (25% upper + 35% lower view), which corresponds to comfortable human visual field [44]. After estimating the center of vision (crossing of the horizon line and viewing direction vertical line, the crop window was set at 1920 × 880 pixels (corresponding to 120:55 ratio) (see Figure 3).

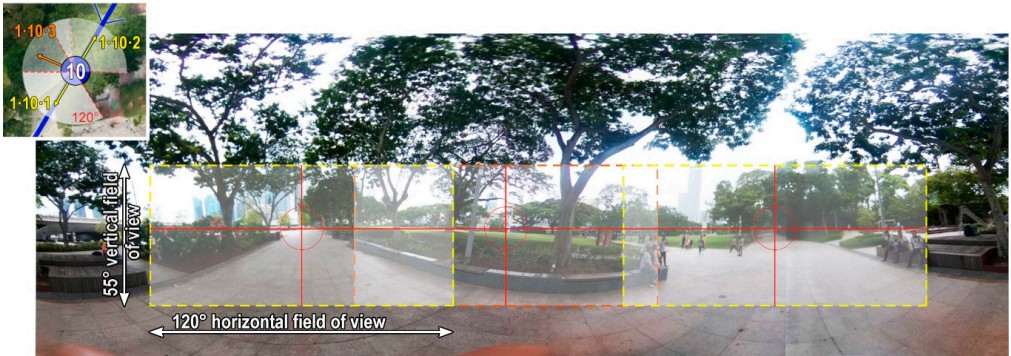

**Figure 3.** Schematic illustration of the 3 viewing angles (1·10·1, 1·10·2 and 1·10·3) extracted from a 360° photo taken at the point 1·10 of the Path 1-Esplanade Park. Yellow and orange dashed lines denote the view limits; red crossing lines denote view's focal point.

In total, 68 assessment points were included (16 at Path 1, 14 at Path 2, 18 at Path 3, and 20 at Path 4), each of which furnished 3 viewing angles, resulting in 204 photos/viewpoints overall to be analyzed.

To facilitate the subsequent activities, a three-digit coding system was adopted, where the first digit corresponded to the path number, the second to the viewing point along the path, and the third to one of the three viewing angles, where 1 refers to the direction along the research route, 2 to the opposite direction, and 3 represents the remaining relevant direction (e.g., avoiding façade of a building we walk along—see Figure 3).

The methodological framework of this study is shown in Figure 4.

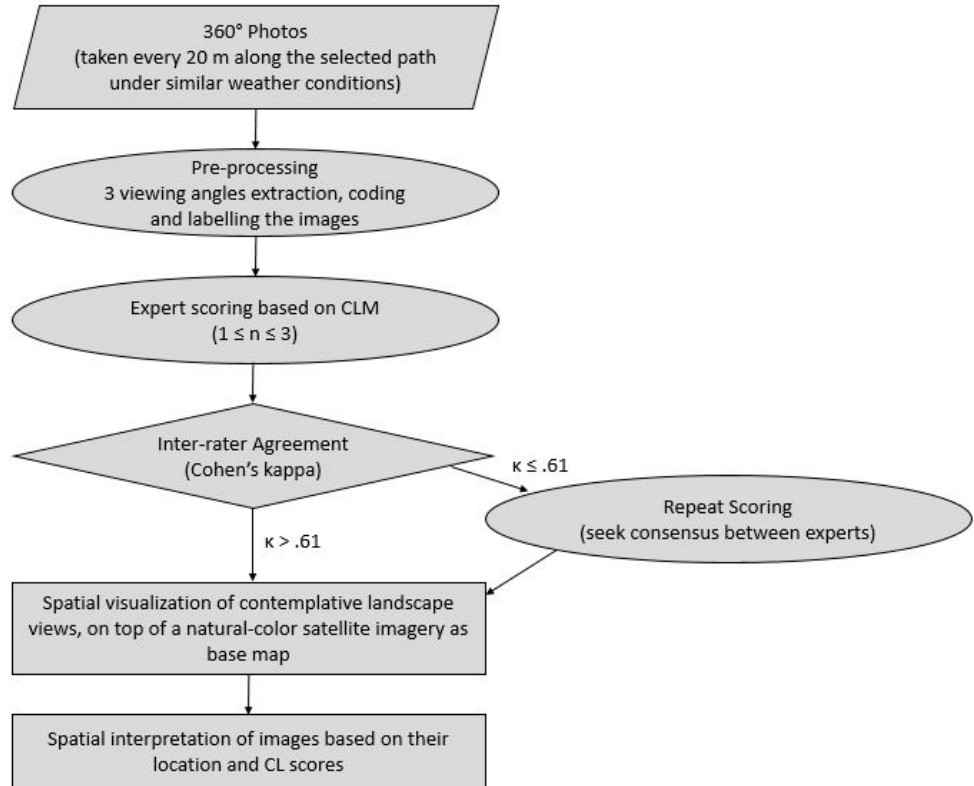

**Figure 4.** Methodology flow chart.

*2.3. Urban Scene Evaluation with CLM*

To ensure objective scoring, each of the 204 coded viewpoints was manually evaluated by three experts [45] trained in landscape architecture according to the 7 key-components of CLM. Each of these

scores was then averaged among three experts and the total contemplative score (CL score) for each view was obtained. The total CL score of the whole path was derived from the average score of each assessment point. To ensure the reliability of the expert assessment, the inter-rater agreement was calculated using the Cohen's kappa measure [46].

*2.4. Spatial Visualization and Analysis*

The experts' scores were then fed into a QGIS software version 3.12.3-București (QGIS Geographic Information System, Open Source Geospatial Foundation Project) for visualization and analysis purposes. Firstly, the maps of the study sites were projected to the EPSG:3857-WGS 84 coordinate system. The projected maps were then georeferenced using Google Maps satellite image imported from online. Finally, a point-based shape-file was created by locating the view-points, followed by inputting the scores associated with each of them to produce an attribute table. The resultant map was then formatted to spatially visualize the CL scores, following the 10-color scale.

## 3. Results

The total agreement between raters across all paths was calculated to be 0.71; the inter-rater agreements of the four paths were 0.83 for Path 1—Esplanade Park, 0.62 for Path 2—Museum Area, 0.62 for Path 3—Robinson Rd., and 0.77 for Path 4—Raffles Place Park, which shows that the experts' assessment reached substantial to perfect value based on Cohen's results classification [46].

*3.1. CL Scores of Scenes*

The highest CL scoring scene was the scene 1·10·1 from the Path 1—Esplanade Park that received 3.33 points (Figure 5a). Conversely, the lowest scoring landscape view was 4·18·1 from the Path 4—Raffles Place Park (Figure 5b). The explanation of the scoring can be found in Section 4.1.

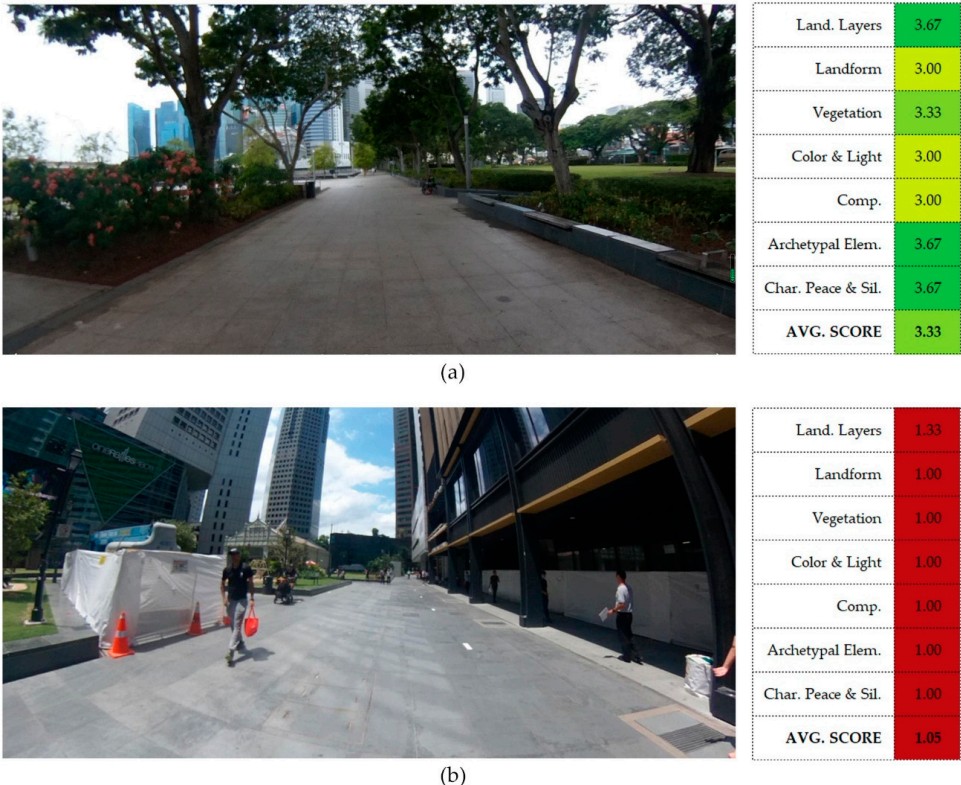

(a)

(b)

**Figure 5.** The highest and the lowest CL scoring scenes of all set with total CL scores and scores for each category, averaged across three experts: (**a**) scene 1·10·1 from *Path 1–Esplanade Park*, with total CL score of 3.33 points and (**b**) scene 4·18·1 from *Path 4–Raffles Place Park*, with CL score of 1.05 points.

### 3.2. CL Scores of Paths

Table 1 presents CL scores for each path averaged across three experts with total CL score and score for each CL category. Path 1—Esplanade Park received the highest overall CL score amongst all paths (2.81 points). With regards to each of the 7 main components of the CLM, Esplanade Park was again given the highest score amongst all the four paths in the evaluation. The lowest scoring paths in terms of contemplative values were Raffles Place Park (1.34 points) and the Robinson Road path (1.54 points), respectively. Neither of the sites' overall scores exceeded the median contemplative score of 3.5 score. However, there were several individual landscape views scoring equal to that value.

**Table 1.** Total CL scores (1-6 point scale) per each of 4 paths and each of the CLM features.

|  | *Path 1*-Esplanade Park | *Path 2*-Museum Area | *Path 3*-Robinson Road | *Path 4*-Raffles Place Park |
|---|---|---|---|---|
| Landscape Layers | 2.89 | 2.02 | 1.73 | 1.46 |
| Landform | 2.24 | 1.76 | 1.38 | 1.10 |
| Vegetation | 2.44 | 1.60 | 1.40 | 1.22 |
| Color & Light | 2.85 | 1.97 | 1.67 | 1.30 |
| Compatibility | 2.76 | 2.11 | 1.88 | 1.57 |
| Archetypal Elements | 3.28 | 1.93 | 1.54 | 1.36 |
| Character of Peace and Silence | 3.25 | 2.07 | 1.18 | 1.35 |
| TOTAL CL SCORE | 2.81 | 1.92 | 1.54 | 1.34 |

### 3.3. Spatial Visualization Map

A spatially explicit maps of CL scores were developed using inverse distance weighted (IDW) interpolation method and enabled determining spatial relationships within and among sites that would have been impossible to discern had the data not been spatialized (selected examples in Figures 6 and 7). For instance, the scenes 3·14~3·16 of Path 3—Robinson Road (Figure 6a–h), are continuously low-scoring scenes. According to the map metadata, these scenes are located in the important entrance square of the large food court, so it is more likely to attract more people; therefore, these areas have the potential to be given priority for design interventions and landscape quality enhancement. Conversely, we can observe continuously medium scoring views between 3·13~3·16 along Path 1—Esplanade Park (Figure 7a–h). The landscape attributes which are contributing to the high scores (in that case visibility of water [Figure 7f] and long-distance views through the bay over the city panorama, [Figure 7a]) are features worth preserving.

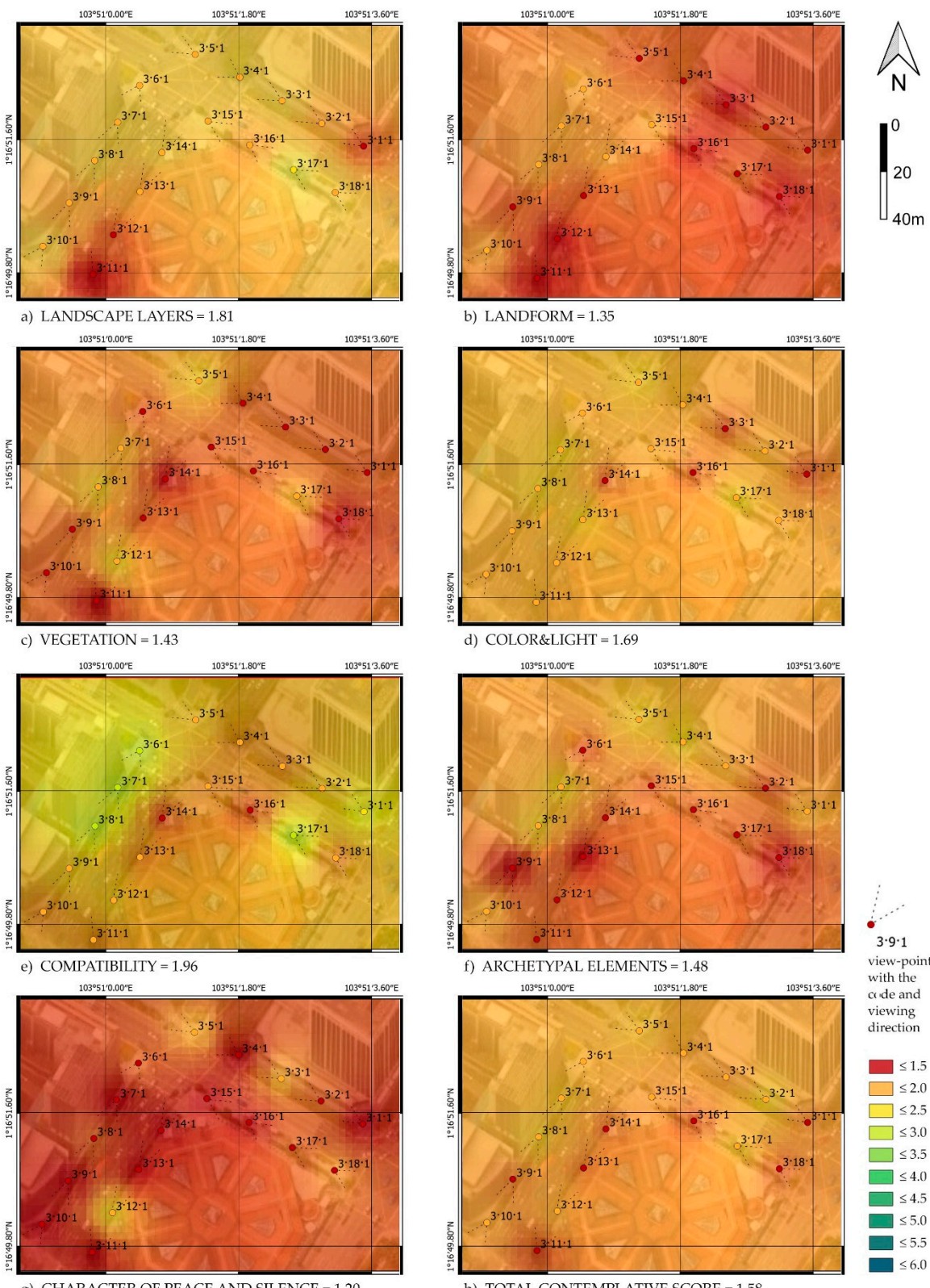

**Figure 6.** Result maps of contemplative scores along the Path 3–Robinson Road; walking in one direction from 3·1 to 3·18; (**a–g**) maps of each contemplative landscape attribute and (**h**) the total CL score (average of all attributes).

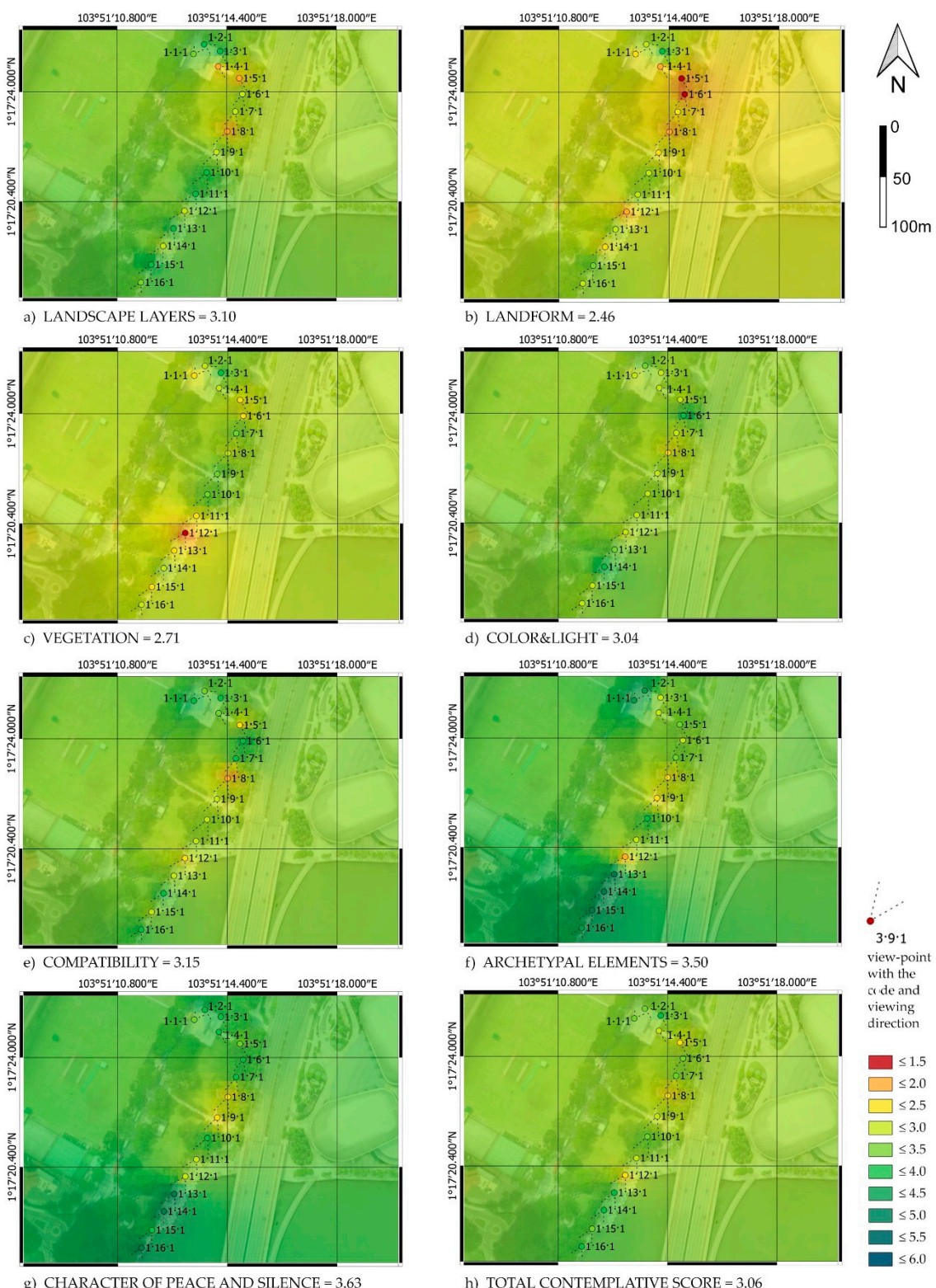

**Figure 7.** Result maps of contemplative scores along the Path 1–Esplanade Park; walking in one direction from 1·1 to 1·16.; (**a–g**) maps of each contemplative landscape attribute and (**h**) the total CL score (average of all attributes).

## 4. Discussion

### 4.1. Summary of Findings, Contributions and Limitations of the Study

This study aimed to test the CLM expert-based tool on the scale of urban walking routes. To this end we examined the contemplativeness of landscape scenes using expert opinions of 204 photos across four popular walking destinations in the Singapore's downtown core— the downtown mainly accommodates working places and offices, hence people spending time there may be more likely to experience stress and burnout. In this study, the CLM was used to assess the individual views and paths, to recommend the design solutions that could improve the quality of views with consideration for the mental health and well-being of urbanites. Testing of the CLM tool in the downtown core area of Singapore was expected to help incorporate this approach in various contexts, such as in urban design and landscape architecture and shed light on the importance of everyday landscape exposures for working professionals.

Our findings demonstrated that even the highest scoring scenes from Path 1—Esplanade Park hardly reached the median score on the CL scale. As the previous research suggests, below 3.5-point scores would not be enough to trigger the brainwave patterns associated with positive mood and restoration [19,22]. This suggests that there is room for design improvements across the study area, especially in terms of landform and vegetation components on which almost all paths scored poorly. The lowest scoring path at Robinson Road was the one characterized by the highest density of infrastructure and built elements and lowest number of natural elements. Owing to CL analysis, the site with the most potential for transformation was identified near the entrance plaza of Telok Ayer Market, and the potential enhancement would target the vegetation, color & light, compatibility, and archetypal elements.

Importing the results into QGIS helped visualize more clearly the total CL scores of views and paths, as well as the broken-down scores of each of the seven key contemplative features in relation to their relative location in space. This yielded an integrated fine-grained spatial output on the contemplativeness of each view. The spatial visualization could help to assess if there is any clustering of similar values. For instance, a continuous juxtaposition of low scores was found in the landform category along Path 4—Raffles Place Park. Since all the data are integrated in spatial format, a proper design intervention could readily be formulated in accordance with the specific attributes that have caused this clustering, so as to fragment this low-score cluster, to potentially mitigate the impact of visual exposures, which may be detrimental to mental health and well-being.

Since the CLM had previously only been applied to single images, the main contribution of this study was therefore extending and testing the framework to assess the contemplativeness of the urban views along walking paths leveraging novel visualization techniques. This, to the best of our knowledge, was the first attempt to map out the areas relevant for human mental health and well-being. The CLM was previously proven to be a reliable and valid psychometric assessment tool in evaluation of single landscape scenes both in situ and in photographic representations [21], but not in larger areas, nor through using 360° photography. The latter was tested and proved robust in this study.

Achieving moderate to perfect inter-rater agreement among experts evaluating the views in this study suggests that only one landscape architecture expert, trained in CLM, would be sufficient to assess the images. However, more studies with multiple experts would be needed to confirm reliability of the scores across various urban contexts and various landscape architecture schools.

To date, there is a paucity of evidence on large-scale relationships between the visual quality of urban scenery and mental health. The most commonly used metrics included the distance between the green space and people's residential address (e.g., [31,47]), and amount of green space area per capita (e.g., [27,48]). More recent works have deemed using the residential address to evaluate environmental exposures as inadequate and an over-simplification of the true state of affairs in view of which other measures have been suggested, such as utilizing wearable tracking devices to uncover true daily routes (e.g., [9]). Another major shortcoming of most of the existing approaches is that the quality

of an exposure is assessed by using satellite imagery, which are by nature incapable of providing a true account of the actual view seen by the human eye. Google Street View may perhaps be more suitable to use. However, the majority of street photos are taken from the top of a car, which may significantly distort the visuo-spatial relations between objects and proportions or landscape elements as compared to the pedestrian point of view. While the overall methodological approach proposed by this study should be applicable to other urban contexts, the main challenge would perhaps be the lack of 360° photos which have to be acquired to enable the use of CLM. Alternatively, the emerging, yet still rather costly, LiDAR scanning method may be a promising tool to acquire realistic quality of exposure data [49].

An operationalized concept of CLM can serve as the basis for creating digital image processing tools that can help evaluate large datasets of photos. Such research is currently underway, for example the prototype of the Contemplative Landscape Automated Scoring System (CLASS), an artificial intelligence client, instantly scores any given digital image of a landscape according to the CLM features with an accuracy comparable to that of a trained expert [50]. There are also emerging automated approaches into landscape imageability focusing of extracting the smallest number of possible viewpoints [51]. Furthermore, research can develop new or enhance existing image processing algorithms to allow development of fully-automated applications. Fully- or semi-automated applications can assist practitioners, such as urban designers and landscape architects, consider the mental health effects of their various developmental scenarios, using 3D visualization [52].

### 4.2. Explanation of Scoring and Design Recommendations

In this section we want to illustrate how our approach using the CLM can lead to practical design solutions to improve the quality of the views and paths. To that end we use the Path 1—Esplanade Park as an example.

The highest scoring scene of the entire set (1·10·1, Figure 5a) was found in Path 1—Esplanade Park. It obtained a CL score of 3.33 points. In this scene, three distinct planes, foreground, mid-ground and the far background can be distinguished in the viewpoint (shrubs in a close-up view, tall trees and CBD towers in a distant view). Nevertheless, they do not explicitly contribute to the overall scene. This explains a relatively high score for the layers of the landscape (3.67 points). In this scene, pedestrians can partly see the diverse skyline through the canopy; the tree-lined alley and the tall buildings in the distance guide the sight. However, because the terrain is flat and lacks natural lines or mounds, the scoring of the landform feature is only at 3 points. The seasonal flowering plants on both sides of the path attract the attention of the visitor. Also, only around six plant species can be distinguished in the scene, giving the impression of moderate biodiversity. Also, the plants seem manicured. These characteristics contributed to the score of 3.33 points received for the vegetation component. Furthermore, lawns and shrubs form a soothing natural green as the dominant color. When sunny, tree shadows will be produced on the ground, displaying a contrast between more and less overlaid shades. The levels of the color & light feature were assessed as moderate, which explains the three-point score for this element. The overall spatial arrangement of the scene is rather harmonious, orderly, and legible. The landscape elements, such as trees, shrubs, seats, squares, and lawns in the scene are physically and visually connected. There are, however, few confusing interference factors in the scene, which caused the scenes to receive a medium score for compatibility (three points). The straight boulevard (path) is a clear archetypal element, which not only guides people's sight, but also dominates the scene. However, the width of the boulevard is making this element less explicitly contributing to the overall scene, therefore the score for archetypal elements in this view was 3.67 points. In this scene, there are shaded seats, which provide people with a comfortable semi-open rest area, the contrast of which with the dense urban core can provide a moderate sense of solitude. This explains the scene receiving a score of 3.67 points in the character of peace and silence category.

Based on the scores for this scene, and also the clustered CL scores for this path, the following design improvements can be recommended for Esplanade Park:

- Introducing natural asymmetry through undulating landforms or natural lines can improve the landform score.
- Introducing a wider variety of plant species, with seasonally changing forms and colors, in the seemingly natural compositions can improve the score for vegetation.
- Changing the width of a path or introducing a different, narrower boulevard pattern can improve the archetypal elements score.
- Providing more comfortable seating with visual divisions to provide a sense of solitude can improve the character of peace and silence score.

Overall, through introducing the above-mentioned design solutions, the overall contemplative score of this site can be improved and the final CL score of the scene can be increased.

## 5. Conclusions

The relationship between daily environmental exposure and multiple mental health issues is now well established. The current mental health crisis, amplified by the COVID-19 pandemic, affecting urbanized areas provokes novel interdisciplinary approaches, and development of new tools to assess the quality of daily environmental exposures. Recognizing the importance of the sensory stimuli we perceive each day for our health and well-being can pave the way to more conscious urban planning and design targeting specific scenery types and components.

This study extended the application of CLM to scales larger than single images by collecting 360° photos of three major walking paths in the downtown core of the compact city of Singapore. The photos taken were then scored by three experts according to the seven contemplative landscape features of the CLM. Our findings demonstrate a high level of consensus among the three experts, pointing out to the potential in developing fully automated procedure. Leveraging novel visualization techniques and spatial mapping, we have shown an approach targeting insights from landscape design and mental health studies towards quality assessment of everyday pedestrian routes. Our CLM-based methodology can inspire planning and design of healthier cities, by providing targeted solutions at the specific site. Likely, many urban cores cannot accommodate major transformations due to land scarcity and the functions they are predestined to serve, but this research shows that applying even small-scale interventions, such as opening the view-sheds, introducing street trees, or removing distracting elements, can elevate the overall contemplative quality of the place.

Future developments should involve building efficient and widely available datasets of street views from the pedestrian perspective, and automated evaluation tools such as CLASS. Further research should investigate the mechanisms underlying the exposures to everyday urban landscapes, with specific focus on longitudinal studies and uncovering the causal relationships between specific contemplative components and the mental health outcomes. Interdisciplinary research should also seek urban design solutions for more personalized self-care interventions.

**Author Contributions:** Conceptualization, H.Y.; methodology, A.O.-G.; software, M.M.; validation, H.Y., M.M., and A.C.; investigation, H.Y.; data curation, H.Y. and A.C.; writing—original draft preparation, H.Y.; writing—review and editing, M.M., A.C., and A.O.-G.; visualization, A.C. and M.M.; supervision, A.O.-G.; funding acquisition, A.O.-G. All authors have read and agreed to the published version of the manuscript.

**Funding:** The article processing charge was paid by NeuroLandscape Foundation.

**Acknowledgments:** Authors would like to thank post-graduate students in Chan Wing Fai for lending the 360° camera for on-site survey and Yu Xi who helped with the site visits in various venues.

**Conflicts of Interest:** The authors declare no conflict of interest.

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
