# Peer review of "Visual Quality Assessment of Urban Scenes with the Contemplative Landscape Model: Evidence from a Compact City Downtown Core"

_remotesensing, doi:10.3390/rs12213517_

Round 1
Reviewer 1 Report
Review of the paper:
Visual Quality Assessment of Urban Scenes with the 2 Contemplative Landscape Model: Evidence from a 3 Compact City Downtown Core
The paper deals with interesting topic related to visual quality of landscape. The chosen theme is interesting especially for planners. The paper is easy to read and understand and well structured. However, I think it is more suitable for a Journal related landscape ecology/landscape planning or GIS.
Reviewer recommends some clarifications:
- Figure 1 present the element for landscape evaluation. The authors have to clarify why the built spaces does not appear in this CLM.
- It is not clear whether the scores were given by local actors or only by experts. What is meant by experts (Delphi expert)?
- Why is this model preferable? There should be at least one comparison with other visual evaluation methods in the section Introduction.
- The research question is missing.
- The discussion section is to much focus on the details of scores.
Authors should discuss more about contributions and limitations of the study in relation to other scientific articles.
E.g.
https://www.sciencedirect.com/science/article/abs/pii/S0169204615000535
- This CLM tool is available online?
Author Response
Reviewer 1
The paper deals with interesting topic related to visual quality of landscape. The chosen theme is interesting especially for planners. The paper is easy to read and understand and well structured. However, I think it is more suitable for a Journal related landscape ecology/landscape planning or GIS.
Response:
Thank you very much for taking the time to pride your kind feedback. We thank the positive assessment of our manuscript, and the journal suggestion as well. We, however, think that the central topic and the methodology presented in the manuscript is as much related to landscape ecology and landscape planning/design as it is to remote sensing since remote sensing has been defined as acquisition of data remotely, without the involvement of physical contact. In our study, the visual quality of landscapes in relation to health impacts was evaluated by the means of 360-degree photographs and rated remotely by the experts, thus qualifies it for the esteemed Journal of Remote Sensing. We particularly think that our research perfectly fits the following special issue: “A Pluralistic Approach to Defining and Measuring Urban Sprawl and its Impacts on Human Well-Being”—we believe that our interdisciplinary approach will be of interest to the wide readership of Remote Sensing.
Reviewer recommends some clarifications:
- Figure 1 present the element for landscape evaluation. The authors have to clarify why the built spaces does not appear in this CLM.
Response:
Thank you for pointing this out. In fact, the CLM does consist of built elements, and the categories of “Compatibility” and “Character of Peace and Silence” are strongly linked with the quality of urban infrastructure, harmony and order of built elements etc. We have also clarified this important point in our manuscript. Please see the changes below.
Line 75 - CLM was developed and operationalized to enable expert-based evaluation of the contemplative value of various views and scenes with special consideration of the urban context, including Urban Green Space (UGS) as built elements are important variables in the Compatibility and Character of Peace and Silence categories
We also updated the caption of Figure 1: Figure 1. Simplified Contemplative Landscape Model for built landscape evaluation with 1 to 6 - point scoring scale; adapted from
- It is not clear whether the scores were given by local actors or only by experts. What is meant by experts (Delphi expert)?
Response:
Thank you for highlighting this flaw. Indeed, we meant the experts in the Delphi sense (an expert may be defined as a person who possesses an expertise in respect of the issue, here landscape architecture: one can acquire this expertise through the general modes such as education, experience, and training). We have added the following reference in the line 221 to clarify what was meant by “expert”:
Hsu, C.-C., & Sandford, B. A. (2007). The Delphi technique: making sense of consensus. Practical Assessment, Research, and Evaluation, 12(1), 10.
- Why is this model preferable? There should be at least one comparison with other visual evaluation methods in the section Introduction.
Response:
Thank you for this comment. To explain the lack of comparative analysis with other models we have added the following statement in the line 75:
CLM is, to the best of our knowledge, the only visual assessment tool employing such a nuanced approach to landscape quality assessment, and which has been operationalised in order to be used in mental health research.
- The research question is missing.
Response.
Thank you. Indeed, in our manuscript, we replaced the research question by the research objectives (lines 134-140). To emphasize this part more, we added a separate sub-chapter “1.3 Research objectives”:
1.3 Research objectives
The goal of this study is to fill this gap in knowledge using the CLM to assess the visual quality of the scenes with consideration to the theory of environmental sensory exposures and their influence on the mental health and well-being of urbanites. We aim to test the CLM tool on the scale of urban walking routes, with larger set images derived from 360° photos taken from the point of view of a pedestrian. It is expected that the outcomes of this study can serve in multiple urban contexts paving the way for more mental-health-aware urban planning and design, and to improvement of the quality of every day environmental exposures.
- The discussion section is to much focus on the details of scores.
Response:
Thank you for pointing this out. We believe that the Reviewer meant the part 4.2 of the Discussion, - Explanation of scoring and design recommendations; where we present how experts arrived at each score of one chosen view. We however feel that this part is very much needed to better illustrate the whole process and potential of the CLM for planning and design practices. In other words, it may provide a guide as to how the CLM could be implemented and utilized in practice. We think it gives additional valuable dimension to our study showcasing and inspiring the different points of view, based on the landscape design evaluation, which for now is only achievable by the human experts (versus machines).
Authors should discuss more about contributions and limitations of the study in relation to other scientific articles.
E.g.
https://www.sciencedirect.com/science/article/abs/pii/S0169204615000535
Response:
Thank you for your kind insight. We cited this very relevant reference in line 331. :
While the overall methodological approach proposed by this study should be applicable to other urban contexts, the main challenge would perhaps be the lack of 360° photos which have to be acquired to enable the use of CLM. Alternatively, the emerging, yet, still rather costly LiDAR scanning method may be a promising tool to acquire realistic quality of exposure data [49].
- This CLM tool is available online?
Response:
Yes, it is added as appendix to a cited paper, which tested its reliability and validity as a psychometric tool. The publication is available online via the following link: https://journals.sagepub.com/doi/abs/10.1177/0265813516660716

Reviewer 2 Report
The enjoyed reading the manuscript, overall the research is well presented and written. However, there is some room for improvement that needs to be addressed.
With regards to the methods please elaborate more on the following:
In lines 191-192 you state "processed in Ricoh Theta Z1 Software UVC4K for Windows (Ricoh 191 Company, Ltd.) and Adobe Photoshop (21.0.3, 2019, Adobe)" what was processed (edited?) in each software?
How were the three angles were defined? (lines 193-194) There is no mention of this. I assume you are referring to the horizontal plane. What angle was the vertical plane? Was this considered?
Furthermore on line 198 you state "represents relevant direction (avoiding facade of a building we walk along)" are you referring to the vertical plane? As above, what angle was the vertical plane? Was this considered?
You mention taking 360o view pictures, following did you crop the images to have 210o views (human visual field), if not why?
In the results sections in Fig. 4. The two images presented have different view angles (vertical plane). Images taken in different view angles are not comparable as the factor view angle influences the score. Please replace one of the image with another image taken with the same view angle.
With regards to the discussion please address the following points:
The application of a method from a large scale landscape to a "small" scale (not to mention different type) would not be 1:1 as scale itself without considering the other factors has it's limitations. You scrape the surface on this (indirectly) but not dirrectly in the discussion area. This limitation is hard to overcome especially on street views. The score of a streetview will always be less than an open view in a park. Also limitations will be greater in narrow streets than wider streets. Based on your method what score should a suggested (optimal) urban streetview have? Can it be achieved, with what improvements?
Author Response
Reviewer 2
The enjoyed reading the manuscript, overall the research is well presented and written. However, there is some room for improvement that needs to be addressed.
Response:
Thank you very much for taking the time to provide us with your kind feedback and comments, which have certainly helped us enhance our manuscript. Please see the response below:
With regards to the methods please elaborate more on the following:
In lines 191-192 you state "processed in Ricoh Theta Z1 Software UVC4K for Windows (Ricoh 191 Company, Ltd.) and Adobe Photoshop (21.0.3, 2019, Adobe)" what was processed (edited?) in each software?
How were the three angles were defined? (lines 193-194) There is no mention of this. I assume you are referring to the horizontal plane. What angle was the vertical plane? Was this considered?
You mention taking 360o view pictures, following did you crop the images to have 210o views (human visual field), if not why?
Response:
Thank you very much for these very important technical comments. We have revised the relevant parts accordingly to further clarify on the methodological aspects; we have also added a figure to better illustrate the process of photo processing (changes are in lines 193-207):
The photos were taken at 165cm above the ground, with a RICOH Theta Z1 360 Camera and pre-processed in Ricoh Theta Z1 Software UVC4K for Windows (Ricoh Company, Ltd.). Three viewpoints were extracted from each photo according to the 120° horizontal and 55° vertical limits (25% upper + 35% lower view), which corresponds to comfortable human visual field [44]. After estimating the center of vision (crossing of the horizon line and walking direction vertical line) the crop window was set at 1920 x 880 pixels (corresponding to 120:55 ratio) (see Figure 3).
Figure 3. Schematic illustration of the 3 viewing angles (1·10·1, 1·10·2 and 1·10·3) extracted from a 360° photo taken at the point 1·10 of the Path 1 - Esplanade Park.
The viewing angle of 210° refers to the full range of the visual field available for a healthy human, that is, including the eyeball movement (without the neck movement), and few degrees pushing it beyond the limit of comfort. This is however not adequate estimation for the passive observation while walking. Anthropometric and ophthalmology literature suggests that the 120° horizontal viewing angle, and 55° vertical angle set the boundary of the “primary field of view” which corresponds to looking straight ahead without eyeball movement, and without including the peripheral vision (linked with the color discrimination). With all this in mind, we set our view limits to 120:55 ratio. Based on this ratio, the single viewing directions were extracted from 360 photos (as presented on the newly added Figure 3).
Furthermore on line 198 you state "represents relevant direction (avoiding facade of a building we walk along)" are you referring to the vertical plane? As above, what angle was the vertical plane? Was this considered?
Response:
We thank for the comment. By the third relevant direction of viewing, we meant any direction other than walking along the path or opposite way. We think that the façade of the building may not be that important of a viewing direction: urban planners, designers, and landscape architects have little to no control to decide about the contents of the façade vistas—such decisions are more up to the developer We, therefore, avoided including the views to the building façade, and only considered other views possible to observe while visiting a given space.
In order to better illustrate where we envision the third relevant direction of possible view would be, we have added Figure 3. We hope that this would help clarify.
In the results sections in Fig. 4. The two images presented have different view angles (vertical plane). Images taken in different view angles are not comparable as the factor view angle influences the score. Please replace one of the image with another image taken with the same view angle.
Response:
Thank you for pointing out our mistake. We have revised our figure accordingly (now it is Figure 5 , line 244)
With regards to the discussion please address the following points:
The application of a method from a large scale landscape to a "small" scale (not to mention different type) would not be 1:1 as scale itself without considering the other factors has it's limitations. You scrape the surface on this (indirectly) but not dirrectly in the discussion area. This limitation is hard to overcome especially on street views. The score of a streetview will always be less than an open view in a park. Also limitations will be greater in narrow streets than wider streets. Based on your method what score should a suggested (optimal) urban Streetview have? Can it be achieved, with what improvements?
Response:
Thank you for your insightful comment. We think what you point out makes an interesting research question; however, we would not be able to afford including this great question since we were more concerned to explore and illustrate the applicability of the CLM to walking paths involving multiple photos in this study. In our Discussion section, however, we have provided possible improvements of the street design (lines 375 – 386), which would improve the contemplative scores of the views. As much as we were tempted to include the simulated new scores, we decided that it would probably be too speculative without validating the simulation. A future study could perhaps take the existing CLM scores, propose improvements, and then create a simulation environment of the improvements in the form of 3D models (or real intervention), and then run the independent expert scoring again and evaluate how the scores have changed.
We do believe that even in the streetscape the contemplative landscape scores can be improved by introducing simple design strategies: openings and closings of the views, introducing vegetation, natural lines and asymmetry. The logic would not be to turn streets into parks, but introduce the compositional features (rather small nuances) that exist in contemplative parks into the urban places that are mostly looked at by people.

Reviewer 3 Report
The authors apply the Contemplative Landscape Model (CLM) to assess the visual quality of some 200 photographs taken from four different paths within the city centre of Singapore. Based on these evaluations the authors demonstrate how to recommend design improvements for the sites. The study fills the gap of existing CLM model by both applying this to urban paths and using 360o photographs. The manuscript contributes to knowledge related to remote imaging, in general, CLM models, in particular. Before the acceptance of the manuscript, I would like to see some editions.
Some specific comments and suggestions are listed below:
- In the Introduction, the authors introduce the CLM model; however, they also need to cite some previous studies which have used the model, summarising their findings and their gaps.
- Line 114 mentions UGS. What is this? Need to be written in the long form.
- Line 131 mentions “the theory of environmental sensory exposures”; however, this has not been explained anywhere. Please specify.
- Line 156: how were the 4 paths selected? based on observations?
- Line 157: “spatial character”, such as??
- Lines 175 and 181: Cross St, Robinson St and the MRT should be labelled on the figures, or else removed. The reader unfamiliar with the site cannot follow.
- Line 190: “site with 20m distance” this is not clear; please clarify. do you mean the distance between each point was 20 m?
- Line 198: “and 3 represents the remaining relevant direction”: not clear what this means? Are not there only 2 walking directions? Maybe an image can help?
- Figure 6: Were the scoring in other viewing directions different? Why only one direction is illustrated? And also, does each scored point include an evaluation of a 360o image?
- There has been emphasis on 360o photography throughout the text, but no images are provided to show 360o photography? For example, Figures 4a and 4b show a photograph shot through a certain viewing angle, not 360o?? These statements should be verified and clarified.
Author Response
Reviewer 3
The authors apply the Contemplative Landscape Model (CLM) to assess the visual quality of some 200 photographs taken from four different paths within the city centre of Singapore. Based on these evaluations the authors demonstrate how to recommend design improvements for the sites. The study fills the gap of existing CLM model by both applying this to urban paths and using 360o photographs. The manuscript contributes to knowledge related to remote imaging, in general, CLM models, in particular. Before the acceptance of the manuscript, I would like to see some editions.
We appreciate the positive evaluation of our manuscript. We have revised our work in accordance with the suggestions and comments. Please see our responses below
Some specific comments and suggestions are listed below:
In the Introduction, the authors introduce the CLM model; however, they also need to cite some previous studies which have used the model, summarising their findings and their gaps.
Thank you for your suggestion. To the best of our knowledge we have cited all CLM studies:
- 1 paper about development and testing of CLM as a psychometric tool (line 72)
- 3 papers considering CLM and neuroscience research ( In Lines 83-90)
- 1 paper on building the image processing software for automated CLM scoring. (line 340)
The relevant description of findings of these studies was described accordingly with the citation.
Line 114 mentions UGS. What is this? Need to be written in the long form.
Response:
Thank you for pointing out our mistake. We have corrected it by spelling it out in Line 75: Urban Green Space (UGS)
Line 131 mentions “the theory of environmental sensory exposures”; however, this has not been explained anywhere. Please specify.
Response:
Thank you for highlighting this. We have described the environmental exposures theory in lines 54-56. In Line 136 we only refer to it again, to better state the objectives of the study.
Line 156: how were the 4 paths selected? based on observations?
Line 157: “spatial character”, such as??
Response:
Thank you for your comment. We have updated the line 161-163 specifying what our selection was based on:
“Four popular walking paths were selected for this study based on the following criteria so that each represents different spatial composition (e.g. enclosed, focal) and viewpoints consisting of various landscape elements (e.g., flat lawn, water body, seating).”
This indeed are based on observation. We also replaced the word “character” by “composition”, and provided some examples.
Lines 175 and 181: Cross St, Robinson St and the MRT should be labelled on the figures, or else removed. The reader unfamiliar with the site cannot follow.
Response:
Thank you for the useful suggestion. We have added the missing labels in the Figure 2.
Line 190: “site with 20m distance” this is not clear; please clarify. do you mean the distance between each point was 20 m?
Response:
Thank you, we indeed overlooked that. This sentence was corrected as follows (line 195):
“…at the shooting points pre-selected before going to the site with 20m distance between each other.”
Line 198: “and 3 represents the remaining relevant direction”: not clear what this means? Are not there only 2 walking directions? Maybe an image can help?
Response:
Thank you for this important comment. We thought our assessment would be more complete if we include possibility of looking sideways (imagine a person walking and turning their heads to the left or to the right). This viewing angle can be also observed from a point of view of person standing on the side of the road, or leaving the building. We think having that 3rd viewing angle is relevant for understanding more about the space, rather than having just opposite angles of viewing.
To better illustrate what we mean by the 3 viewing direction we have added Figure 3 (line 202).
Figure 6: Were the scoring in other viewing directions different? Why only one direction is illustrated? And also, does each scored point include an evaluation of a 360o image?
Response:
Thank you for your comment on this Figure. Yes, the scoring is different with each direction of the viewing. In fact, we only selected a small portion of the data to produce the Figure 6. Because of space limitation, we chose to present only one viewing direction scores. The purpose is simply to illustrate to the reader how this looks like.
Each point illustrates score obtained by evaluation of only 1 viewing direction (120°), as indicated in the Figure 6 (and a new Figure 7).
We also updated the captions for Figure 6 and 7, highlighting specifically the only one walking direction:
Line 270: Figure 6. Result maps of scores of each contemplative landscape attribute as well as the total CL score along the Path 3 – Robinson Road; walking in one direction from 3·1 to 3·18.
Line 274: Figure 7. Result maps of scores of each contemplative landscape attribute as well as the total CL score along the Path 1 – Esplanade Park; walking in one direction from 1·1 to 1·16.
There has been emphasis on 360o photography throughout the text, but no images are provided to show 360o photography? For example, Figures 4a and 4b show a photograph shot through a certain viewing angle, not 360o?? These statements should be verified and clarified.
Response:
We appreciate your comment. According to suggestion, we have added Figure 3 as an example of 360° picture, illustrating how we extracted 3 views from 360° photograph. Please take note that displaying 360° photos on a flat page results in certain level of image distortion.
Round 2
Reviewer 1 Report
The authors made insignificant changes:!!!
rows 77-79 - a detail of the model, un short paragraph
Figure 3 - is to be appreciated
BUT I do not find the changes/completions requested by the referent.
If they have made other changes, mark them in the text.see below
The reviewer recommends some clarifications:
- Figure 1 presents the element for landscape evaluation. The authors have to clarify why the built spaces do not appear in this CLM.
- It is not clear whether the scores were given by local actors or only by experts. What is meant by experts (Delphi expert)?
- Why is this model preferable? There should be at least one comparison with other visual evaluation methods in section Introduction.
- The research question is missing.
- The discussion section is to much focus on the details of scores.
Authors should discuss more the contributions and limitations of the study in relation to other scientific articles.
E.g.
https://www.sciencedirect.com/science/article/abs/pii/S0169204615000535
- This CLM tool is available online?
Author Response
Dear Reviewer,
Thank you for your comments.
We think the track changes in the revised manuscript were not set properly to display all changes. Only some small changes were displayed in colour.
We attach the pdf version with all of our changes answering reviewers comments marked in red.
Additionally, I repeat below the answer to queries 1-6, for easier reference.
1. Figure 1 presents the element for landscape evaluation. The authors have to clarify why the built spaces does not appear in this CLM.
Response:
Thank you for pointing this out. In fact, the CLM does consist of built elements, and the categories of “Compatibility” and “Character of Peace and Silence” are strongly linked with the quality of urban infrastructure, harmony and order of built elements etc. We have also clarified this important point in our manuscript. Please see the changes below.
Line 75 - CLM was developed and operationalized to enable expert-based evaluation of the contemplative value of various views and scenes with special consideration of the urban context, including Urban Green Space (UGS) as built elements are important variables in the Compatibility and Character of Peace and Silence categories
We also updated the caption of Figure 1: Figure 1. Simplified Contemplative Landscape Model for built landscape evaluation with 1 to 6 - point scoring scale; adapted from [22].
2. It is not clear whether the scores were given by local actors or only by experts. What is meant by experts (Delphi expert)?
Response:
Thank you for highlighting this flaw. Indeed, we meant the experts in the Delphi sense (an expert may be defined as a person who possesses an expertise in respect of the issue, here landscape architecture: one can acquire this expertise through the general modes such as education, experience, and training). We have added the following reference in the line 221 to clarify what was meant by “expert”:
Hsu, C.-C., & Sandford, B. A. (2007). The Delphi technique: making sense of consensus. Practical Assessment, Research, and Evaluation, 12(1), 10.
3. Why is this model preferable? There should be at least one comparison with other visual evaluation methods in the section Introduction.
Response:
Thank you for this comment. To explain the lack of comparative analysis with other models we have added the following statement in the line 75:
CLM is, to the best of our knowledge, the only visual assessment tool employing such a nuanced approach to landscape quality assessment, and which has been operationalised in order to be used in mental health research.
4. The research question is missing.
Response.
Thank you. Indeed, in our manuscript we replaced the research question by the research objectives (lines 134-140). To emphasize this part more, we added a separate sub-chapter “1.3 Research objectives”:
1.3 Research objectives
The goal of this study is to fill this gap in knowledge using the CLM to assess the visual quality of the scenes with consideration to the theory of environmental sensory exposures and their influence on the mental health and well-being of urbanites. We aim to test the CLM tool on the scale of urban walking routes, with larger set images derived from 360° photos taken from the point of view of a pedestrian. It is expected that the outcomes of this study can serve in multiple urban contexts paving the way for more mental-health-aware urban planning and design, and to improvement of the quality of every day environmental exposures.
5. The discussion section is to much focus on the details of scores.
Response:
Thank you for pointing this out. We believe that the Reviewer meant the part 4.2 of the Discussion, - Explanation of scoring and design recommendations; where we present how experts arrived at each score of one chosen view. We however feel that this part is very much needed to better illustrate the whole process and potential of the CLM for planning and design practices. In other words, it may provide a guide as to how the CLM could be implemented and utilized in practice. We think it gives additional valuable dimension to our study showcasing and inspiring the different points of view, based on the landscape design evaluation, which for now is only achievable by the human experts (versus machines).
Authors should discuss more about contributions and limitations of the study in relation to other scientific articles.
E.g.
https://www.sciencedirect.com/science/article/abs/pii/S0169204615000535
Response:
Thank you for your kind insight. We cited this very relevant reference in line 331. :
While the overall methodological approach proposed by this study should be applicable to other urban contexts, the main challenge would perhaps be the lack of 360° photos which have to be acquired to enable the use of CLM. Alternatively, the emerging, yet, still rather costly LiDAR scanning method may be a promising tool to acquire realistic quality of exposure data [49].
6. This CLM tool is available online?
Response:
Yes, it is added as appendix to a cited paper, which tested its reliability and validity as a psychometric tool. The publication is available online via the following link: https://journals.sagepub.com/doi/abs/10.1177/0265813516660716

Reviewer 2 Report
Your manuscript has been improved and I have no further comments.
Author Response
Thank you for your positive feedback!